# Dysregulation of MicroRNAs in Hepatocellular Carcinoma: Targeting Oncogenic Signaling Pathways for Innovative Therapies

**DOI:** 10.3390/ijms26178365

**Published:** 2025-08-28

**Authors:** Yusra Zarlashat, Judit Halász, Edit Dósa

**Affiliations:** 1Department of Biochemistry, Government College University Faisalabad, Faisalabad 38000, Pakistan; yusrazarlashat@gcuf.edu.pk; 2Department of Pathology, Forensic and Insurance Medicine, Semmelweis University, 1091 Budapest, Hungary; 3Heart and Vascular Center, Semmelweis University, 1122 Budapest, Hungary; dosa.edit@semmelweis.hu

**Keywords:** hepatocellular carcinoma, microRNA, oncomiR, tumor suppressor microRNA, signaling pathways, biomarkers, therapeutic targets

## Abstract

Hepatocellular carcinoma (HCC) is one of the most common malignancies worldwide and the third leading cause of cancer-related death. Hyperactivation of oncogenes and suppression of tumor suppressor genes/proteins drive HCC initiation and progression. MicroRNAs (miRNAs) critically modulate HCC biology by regulating proliferation, apoptosis, and metastasis. Acting either as tumor suppressors or oncomiRs, they shape core signaling pathways, including PI3K/Akt/mTOR, Hippo–YAP/TAZ, Wnt/β-catenin, RAS/MAPK, and p53. Their dysregulation in tissues and body fluids renders them promising diagnostic biomarkers and therapeutic targets. Preclinical studies demonstrate that miRNA-based strategies—either restoring tumor-suppressive miRNAs (e.g., miR-34a, miR-125a-5p) or inhibiting oncogenic miRNAs (e.g., miR-660-5p)—can suppress HCC progression and reduce treatment resistance. Combination approaches, such as pairing miR-122 mimics with miR-221 inhibitors or delivering miR-326 via nanoparticles, further enhance efficacy by simultaneously targeting multiple oncogenic pathways. This review summarizes recent advances in miRNA-mediated regulation of HCC signaling and highlights their clinical potential, including ongoing trials of miRNA-based diagnostics and therapeutics for early detection, prognostication, and personalized treatment.

## 1. Introduction

Hepatocellular carcinoma (HCC) is the most common primary liver cancer and the third leading cause of cancer-related mortality worldwide [1]. A complex network of proteins orchestrates hepatic cellular functions, and imbalances in these networks disrupt normal liver physiology, thereby driving tumorigenesis. Aberrant activation of oncoproteins or suppression of tumor suppressor proteins leads to unchecked proliferation, impaired differentiation, and resistance to apoptosis [2]. Chronic liver injury and cirrhosis, often related to alcohol consumption, viral hepatitis, metabolic dysfunction-associated steatotic liver disease, and genetic predisposition, are major contributors to HCC development [3].

Despite advances in treatment, substantial diagnostic and therapeutic challenges remain. More than 70% of cases are diagnosed at advanced stages because of inadequate surveillance and non-specific early symptoms, which limit curative options. Transarterial chemoembolization and radioembolization can control disease progression but are not curative. Targeted agents, anti-angiogenic therapies, and immunotherapies are promising, yet their benefits are tempered by tumor heterogeneity, immune evasion, and only modest survival gains [1]. The absence of reliable biomarkers also constrains patient stratification and contributes to poor outcomes, underscoring the need for improved strategies [4].

MicroRNAs (miRNAs) are small, single-stranded, non-coding RNAs (18–24 nucleotides) that regulate gene expression post-transcriptionally by binding to target messenger RNAs (mRNAs) and inducing degradation or translational repression. By modulating both oncogenic and tumor-suppressive pathways, miRNAs influence HCC progression and represent attractive therapeutic targets [5]. Importantly, miRNAs are stable and readily detectable in body fluids, particularly saliva [6], blood [7], and urine [8], supporting their utility as biomarkers for diagnosis and prognosis [9]. Dysregulated miRNAs promote hepatic tumorigenesis by post-transcriptionally regulating tumor suppressors and oncogenes, thereby altering signaling pathways that govern proliferation, differentiation, and apoptosis [10]. Among these pathways, phosphatidylinositol 3-kinase (PI3K)/protein kinase B (Akt)/mammalian target of rapamycin (mTOR) signaling is frequently dysregulated in HCC and plays a central role in tumor development, angiogenesis, metastasis, and chemoresistance [11]. The Hippo–Yes-associated protein (YAP)/transcriptional co-activator with PDZ-binding motif (TAZ) pathway, which is essential for controlling cell growth and death, is often disrupted [12]. Likewise, dysregulation of the Wnt/β-catenin pathway—a key regulator of liver growth and repair—promotes cancer dissemination and treatment resistance [13]. The RAS/mitogen-activated protein kinase (MAPK) pathway is frequently overactive, supporting cancer cell survival and proliferation [14]. Finally, impairment of the p53 tumor suppressor pathway, which orchestrates cellular responses to stress, including DNA damage, is a common feature of HCC [15]. Given their central roles in these pathways, miRNAs are compelling candidates for therapy, and synthetic miRNA mimics have already entered clinical testing. However, challenges remain, including off-target effects and limited delivery efficiency. To address these limitations, innovative delivery systems, such as lipid nanoparticles (LNPs), viral vectors, and engineered exosomes, are being developed to enhance tumor specificity and minimize adverse effects. In parallel, miRNA-based combination strategies are being explored to overcome drug resistance and improve efficacy. This review examines how miRNAs regulate HCC signaling and highlights translational opportunities for earlier diagnosis, improved response prediction, and better patient outcomes.

## 2. miRNA Biogenesis and Regulation in HCC

The first step in miRNA synthesis is the transcription of the primary transcript (pri-miRNA), which is 3′ polyadenylated and 5′ capped. Transcription is usually mediated by RNA polymerase II, although RNA polymerase III can also transcribe certain pre-miRNAs. The microprocessor complex, composed of the RNA-binding protein DGCR8 (DiGeorge syndrome critical region 8) and the type III RNase DROSHA, then cleaves pri-miRNAs into precursor miRNAs (pre-miRNAs), ~85-nt stem-loop structures [16]. In HCC, DROSHA and DICER are often downregulated, contributing to global miRNA dysregulation and aggressive tumor behavior [17]. After nuclear export via the RanGTP (RAS-related nuclear protein)/Exportin-5 complex, DICER (an RNase III enzyme) processes pre-miRNAs into ~20–22-nt miRNA/miRNA duplexes [16]. In HCC, suppression of Exportin-5 impairs the export of miR-122—a liver-specific tumor suppressor—whose loss promotes metabolic dysfunction and chemoresistance. Upon strand separation, the mature miRNA is loaded into the RNA-induced silencing complex (RISC), which is subsequently guided to target mRNAs [18]. Target recognition is primarily mediated by the 6–8-nt seed region at the 5′ end of the miRNA [19]. Depending on the degree of complementarity, RISC either triggers mRNA degradation (near-perfect pairing) or inhibits translation (imperfect pairing) (Figure 1).

## 3. PI3K/Akt/mTOR Pathway

The PI3K/Akt/mTOR signaling pathway is a central regulator of cellular processes and is frequently dysregulated in HCC, driving tumor progression, metabolic reprogramming, and therapeutic resistance [20]. This pathway is initiated when growth factors, such as epidermal growth factor (EGF) or insulin-like growth factor 1 (IGF-1), bind to receptor tyrosine kinases. This interaction triggers PI3K to convert phosphatidylinositol 4,5-bisphosphate (PIP2) into phosphatidylinositol 3,4,5-trisphosphate (PIP3) at the plasma membrane, a critical step that recruits and activates Akt [21]. miRNAs modulate the PI3K/Akt/mTOR pathway in HCC either as oncomiRs or tumor suppressor miRNAs, thereby influencing tumor progression, metastasis, and treatment responses. They exert their effects by repressing tumor suppressor genes or disrupting translational regulation, ultimately permitting oncogenic signaling [22] (Figure 2A).

Oncogenic miRNAs: The metastasis-associated in colon cancer 1 (MACC1) gene promotes invasion, metastasis, and proliferation in HCC through activation of c-Met signaling. c-Met dimerization, stabilized by MACC1, enhances PI3K recruitment via GRB2 (growth factor receptor-bound protein 2) adaptor proteins, leading to Akt/mTOR pathway activation. Zhang et al. demonstrated that miR-34a and miR-125a-5p suppress HCC progression by targeting MACC1. These miRNAs inhibit proliferation, metastasis, and tumor growth while promoting apoptosis through inhibition of PI3K/Akt signaling [23]. Notably, miR-34a also directly targets the 3′UTR of mesenchymal–epithelial transition factor (MET) mRNA, exerting a dual inhibitory effect on this oncogenic pathway. miR-93 has also been identified as an oncogenic miRNA in HCC. It promotes tumor progression by directly suppressing phosphatase and tensin homolog (PTEN) and cyclin-dependent kinase inhibitor 1A (CDKN1A), thereby enhancing c-Met/PI3K/Akt signaling. High miR-93 expression correlates with poor prognosis, increased proliferation, migration, invasion, and reduced apoptosis. Interestingly, it also enhances sensitivity to sorafenib and tivantinib, suggesting therapeutic relevance [24]. miR-660-5p promotes proliferation, invasion, colony formation, and tumorigenicity of HCC cells, as confirmed in both in vitro and in vivo studies. Its downregulation, by contrast, inhibits cancer cell growth. Mechanistically, miR-660-5p directly targets YWHAH (tyrosine 3-monooxygenase/tryptophan 5-monooxygenase activation protein, eta 14-3-3η), a signaling adaptor that facilitates PI3K/Akt activation, thereby inducing epithelial–mesenchymal transition (EMT) [25]. Consistently, high miR-660 expression in tumor tissues and cell lines increases the proportion of cells in the S phase, while its suppression reduces proliferation. The miR-660-5p/YWHAH axis thus drives proliferation, migration, invasion, EMT, and tumorigenesis in HCC. During EMT, miR-660-5p-mediated Akt activation expels forkhead box O (FOXO) transcription factors from the nucleus. FOXO inactivation derepresses EMT-promoting genes such as SNAIL (Snail family transcriptional repressors) and TWIST (twist family bHLH transcription factors) [26]. miR-3691-5p is another oncomiR upregulated in HCC tissues and cells. It promotes tumor progression by downregulating PTEN, a lipid phosphatase that converts PIP3 back to PIP2 and thus prevents excessive Akt activation. Loss of PTEN activity removes this regulatory brake, enabling unchecked proliferation [27].

Tumor suppressor miRNAs: miR-30b-3p is significantly downregulated in HCC tissues, with higher expression correlating with better overall survival. It inhibits cell viability, proliferation, migration, and invasion by directly targeting tripartite motif-containing 27 (TRIM27), an E3 ubiquitin ligase that promotes PI3K/Akt activation. TRIM27 destabilizes PTEN transcripts by degrading PTEN mRNA-binding proteins. Loss of miR-30b-3p, therefore, increases TRIM27 levels, accelerates PTEN mRNA decay, and relieves suppression of the PI3K/Akt pathway. The miR-30b-3p/TRIM27/PI3K/Akt axis thus plays a crucial role in HCC progression and may serve as a biomarker for diagnosis and therapy [28]. miR-133b also acts as a tumor suppressor. Its low expression is associated with poor prognosis after surgery, whereas overexpression inhibits HCC growth, induces apoptosis, and suppresses EGF receptor (EGFR)/PI3K/Akt/mTOR signaling. These findings highlight miR-133b as both a prognostic marker and a therapeutic candidate [29]. miR-1914 functions as a tumor suppressor by inducing cell-cycle arrest and apoptosis in HCC cells. Its direct target is GPR39, a zinc-activated G protein-coupled receptor that negatively regulates PI3K/Akt/mTOR signaling [30]. When GPR39 is activated, it engages Gαq proteins that inhibit PI3K (p110 subunit), thereby attenuating pathway activation.

Given the extensive regulatory role of miRNAs in PI3K/Akt-driven hepatocarcinogenesis, miRNA-based interventions represent a promising strategy for HCC management. Table 1 summarizes several miRNAs implicated in the PI3K/Akt/mTOR pathway and highlights their roles in proliferation, invasion, metastasis, and therapeutic potential as diagnostic biomarkers or treatment targets.

## 4. Hippo–YAP/TAZ Pathway

The Hippo–YAP/TAZ signaling pathway is a crucial regulatory mechanism activated by mechanical cues such as cell–cell interactions, extracellular ligands, and cellular receptors. It modulates cell adhesion, polarity, and other essential processes fundamental for development and tissue homeostasis. Dysregulation of this pathway is closely associated with HCC, where abnormal YAP/TAZ activation drives tumor development, metastasis, and therapy resistance [12]. YAP and TAZ are transcriptional co-activators that directly regulate miRNA synthesis by binding to promoters of miRNA genes or by modulating the expression of miRNA-processing machinery. Conversely, several miRNAs regulate Hippo pathway components, establishing feedback loops [37] (Figure 2B).

Oncogenic miRNAs: miR-21 is an oncomiR that regulates Hippo signaling through multiple axes. By targeting YOD1, a deubiquitinase that stabilizes the E3 ubiquitin ligase ITCH, it indirectly suppresses large tumor suppressor kinase (LATS) and activates YAP/TAZ, enhancing liver growth and tumor progression [38]. In addition, miR-21-3p directly targets SMAD7 (mothers against decapentaplegic homolog 7), a negative regulator of transforming growth factor beta (TGF-β) signaling, thereby enhancing YAP1 activity. High miR-21-3p levels correlate with poor prognosis, while SMAD7 restoration partly reverses its oncogenic effects [39]. miR-135b is upregulated in HCC and promotes tumor progression by suppressing mammalian sterile 20-like kinase 1 (MST1), thereby inhibiting Hippo signaling. MST1 normally phosphorylates and activates LATS1/2 kinases, which in turn phosphorylate YAP/TAZ to prevent their nuclear translocation. When MST1 is inhibited by miR-135b, LATS1/2 remain inactive, allowing YAP/TAZ to persist in the nucleus and drive oncogenic transcription. High miR-135b levels correlate with advanced disease and poor survival, making it a potential prognostic biomarker and therapeutic target [40]. The hypoxic tumor microenvironment also stimulates Hippo dysregulation. Hypoxia-inducible factors (HIF-1α/2α) directly bind the miR-512-3p promoter, increasing its expression under low oxygen conditions. miR-512-3p targets LATS2, preventing formation of the active LATS2–Mps One Binder 1 (MOB1) complex required for YAP phosphorylation. Consequently, YAP remains active, promoting proliferation and migration in HCC. High miR-512-3p levels associate with poor clinical features, including advanced tumor–node–metastasis stage [41]. miR-1307-3p also promotes HCC growth. It is activated via Meis homeobox 2D (MEIS2D) in cooperation with pre-B-cell leukemia transcription factor 1 (PBX1) and directly targets LATS1, reducing YAP phosphorylation. Loss of this tumor-suppressive checkpoint results in YAP activation [42]. Elevated miR-1307-3p levels correlate with venous infiltration, larger tumor size, advanced stages, and poor prognosis. Its knockdown suppresses proliferation, migration, and invasion in vitro and in vivo, underscoring its potential as a therapeutic target [43].

Tumor suppressor miRNAs: In contrast, several miRNAs act as tumor suppressors in Hippo signaling. miR-29c-3p inhibits HCC tumorigenesis by repressing DNA methyltransferase 3B (DNMT3B), thereby preventing hypermethylation and silencing of tumor suppressor genes. At the same time, it binds the 3′UTR of LATS1, influencing Hippo regulation. Overexpression of miR-29c-3p suppresses migration, proliferation, and tumor growth in vivo, making it a candidate therapeutic mimic [44]. miR-375, often downregulated due to epigenetic silencing, targets YAP1, TEA domain transcription factor 4 (TEAD4), and connective tissue growth factor (CTGF). Low miR-375 levels associate with poor prognosis and metastasis. Restoring its expression inhibits tumor growth, highlighting the therapeutic potential of miR-375 mimics or Hippo pathway inhibitors [45].

Taken together, these studies underscore the complex regulatory network between miRNAs and the Hippo–YAP/TAZ pathway in HCC. Therapeutic strategies may exploit this interplay by developing antagomiRs against oncogenic miRNAs (e.g., miR-21/miR-21-3p, miR-135b, miR-1307-3p) or mimics for tumor suppressor miRNAs (e.g., miR-29c-3p, miR-375). A selection of key miRNAs regulating the Hippo pathway in HCC is summarized in Table 2.

## 5. Wnt/β-Catenin Pathway

The Wnt/β-catenin signaling pathway contributes to tumor initiation and progression by promoting uncontrolled proliferation, survival, and metastasis [47]. In the absence of Wnt signals, β-catenin is phosphorylated by a destruction complex (axis inhibition protein [AXIN1/2], adenomatous polyposis coli [APC], glycogen synthase kinase 3 beta [GSK3β], and casein kinase 1 alpha [CK1α]) and subsequently degraded by the proteasome. This critical regulatory mechanism is frequently disrupted in HCC [48]. miRNAs, functioning either as tumor suppressors or oncomiRs, modulate key components of the Wnt/β-catenin pathway, thereby influencing metastasis, invasion, proliferation, and therapy resistance in HCC [13] (Figure 2C). When Wnt ligands bind Frizzled/low-density lipoprotein receptor-related protein (LRP) receptors, the destruction complex is inhibited, allowing β-catenin to accumulate and translocate into the nucleus. There, it activates TCF (T-cell factor)/LEF (lymphoid enhancer-binding factor) transcription factors, driving the expression of proliferative genes such as c-MYC and cyclin D1 (CCND1). The tumor suppressor WT1 (Wilms tumor 1) also modulates Wnt activity in a context-dependent manner, influencing hepatic development, vascular formation, and hepatocarcinogenesis.

Oncogenic miRNAs: Several miRNAs activate the Wnt/β-catenin pathway in HCC. miR-19a-3p and miR-376c-3p promote Wnt activation by targeting SRY-box transcription factor 6 (SOX6), a factor that normally restrains β-catenin at the membrane. Their inhibition of SOX6 releases β-catenin for nuclear translocation, facilitating HCC progression [49]. miR-197 targets AXIN2, naked cuticle homolog 1 (NKD1), and Dickkopf-related protein 2 (DKK2), thereby activating Wnt/β-catenin signaling and inducing EMT and invasion; high miR-197 levels are associated with portal vein metastasis [50]. miR-500a directly suppresses GSK3β, preventing β-catenin degradation and promoting migration and invasion. Elevated miR-500a expression correlates with poor prognosis [51]. miR-1246 enhances stemness, drug resistance, and metastasis of liver cancer stem-cells by suppressing AXIN2 and GSK3β. This stabilization of β-catenin is driven by an octamer-binding transcription factor 4 (OCT4)/miR-1246 axis and correlates with poor prognosis [52].

Tumor suppressor miRNAs: Conversely, multiple miRNAs inhibit Wnt signaling in HCC. miR-194 targets protein regulator of cytokinesis 1 (PRC1), a stabilizer of β-catenin. By inhibiting PRC1, miR-194 suppresses EMT, invasion, and migration. miR-194 is downregulated in HCC, while PRC1 is upregulated, and restoring miR-194 impedes tumor growth [53]. miR-329-3p suppresses ubiquitin-specific peptidase 22 (USP22), a deubiquitinase that stabilizes β-catenin, thereby reducing β-catenin half-life and limiting proliferation and migration. Low miR-329-3p expression correlates with HCC progression [54]. miR-342 targets CXC motif chemokine 12 (CXCL12), disrupting the CXCL12/CXC chemokine receptor type 4 (CXCR4) loop that sustains β-catenin activity in the tumor microenvironment, thus suppressing proliferation and enhancing apoptosis [55]. miR-361-5p exerts dual inhibitory effects by downregulating mesenchymal markers (N-cadherin, vimentin) and targeting TWIST1, a transcription factor promoting EMT [56]. miR-361-5p restoration suppresses proliferation, migration, invasion, and metastasis, positioning it as a promising therapeutic biomarker [57]. Additional evidence highlights miR-409-3p as a tumor suppressor; it targets the oncogene BRF2 (transcription factor IIIB), whose overexpression promotes invasion and metastasis. Bioinformatics analyses suggest BRF2 acts via Wnt/β-catenin signaling, linking miR-409-3p to early HCC regulation [58]. Finally, miR-485-5p inhibits WW domain-binding protein 2 (WBP2), a co-activator that enhances β-catenin/TCF4 transcriptional activity, thereby blocking proliferation, migration, and invasion [59].

Collectively, these findings highlight the pivotal role of miRNAs in modulating Wnt/β-catenin signaling in HCC. Oncogenic miRNAs such as miR-197, miR-500a, and miR-1246 activate β-catenin signaling and drive EMT, stemness, and metastasis. In contrast, tumor-suppressive miRNAs such as miR-329-3p, miR-342, and miR-361-5p inhibit pathway activity, restoring control over proliferation, apoptosis, and migration. Therapeutic approaches may involve antagomiRs targeting oncomiRs (e.g., miR-19a-3p, miR-1246) or mimics of tumor suppressors (e.g., miR-342, miR-361-5p), potentially in combination with pharmacological Wnt inhibitors. Table 3 summarizes key miRNAs regulating the Wnt/β-catenin pathway in HCC.

## 6. RAS/MAPK Pathway

The RAS/MAPK signaling pathway is frequently dysregulated in HCC, largely through overactivation of receptor tyrosine kinases (RTKs) [63]. Upon growth factor (EGF, hepatocyte growth factor [HGF], IGF) binding, RTKs recruit adaptor proteins (GRB2–Son of Sevenless [SOS]), which activate RAS GTPase—a key molecular switch that initiates downstream signaling. Activated RAS recruits RAF kinases (A-Raf proto-oncogene, serine/threonine kinase [ARAF], B-Raf proto-oncogene, serine/threonine kinase [BRAF], C-Raf proto-oncogene, serine/threonine kinase [CRAF]), which then phosphorylate MAP/ERK kinases 1 and 2 (MEK1/2). In turn, MEK1/2 activate extracellular signal-regulated kinases 1 and 2 (ERK1/2), terminal kinases that translocate to the nucleus to regulate transcription factors such as c-Fos and c-Jun [64]. Dysregulation of this cascade enhances proliferation, survival, and invasion, and is closely interconnected with the PI3K/Akt/mTOR and Wnt/β-catenin pathways, promoting both HCC progression and therapy resistance [65]. miRNAs influence this pathway either as tumor suppressors or oncomiRs, thereby shaping hepatocarcinogenesis [14] (Figure 2D).

Oncogenic miRNAs: Several miRNAs act as oncomiRs by activating MAPK signaling. miR-126-3p directly suppresses sprouty-related EVH1 domain-containing protein 1 (SPRED1), an inhibitor of ERK signaling. High miR-126-3p expression promotes sorafenib resistance, while its inhibition restores drug sensitivity by upregulating SPRED1 [66]. miR-330-5p targets sprouty RTK signaling antagonist (SPRY2), a negative regulator that normally interacts with GRB2 to block RAS activation. Its suppression by miR-330-5p sustains MAPK activation, promoting proliferation and tumor growth. High levels correlate with poor prognosis [67].

Tumor suppressor miRNAs: In contrast, several miRNAs inhibit RAS/MAPK signaling and suppress HCC progression. miR-30a is downregulated in metastatic HCC, where it suppresses Beclin 1- and autophagy protein 5 (ATG5)-dependent autophagy. Its loss correlates with vascular invasion, recurrence, and poor prognosis, while restoration blocks autophagy-mediated metastasis [68]. miR-101 forms a reciprocal negative feedback loop with enhancer of zeste homolog 2 (EZH2). EZH2 epigenetically represses miR-101 transcription, while miR-101 inhibits EZH2 expression. Both miR-101 restoration and EZH2 silencing reduce proliferation and metastasis, underscoring their coordinated tumor-suppressive role [69]. miR-122, a liver-specific tumor suppressor, directly targets IGF-1R. By reducing IGF-1R signaling, miR-122 suppresses both MAPK and PI3K/Akt pathways, thereby limiting proliferation and promoting apoptosis [70]. miR-148a-3p is reduced in HCV-related HCC and directly targets c-Jun, thereby limiting MAPK-driven proliferation and promoting apoptosis. Its downregulation enhances tumor growth, while its restoration inhibits HCC progression [71]. miR-203 directly targets neuroblastoma RAS viral oncogene homolog (NRAS), thereby inactivating MAPK signaling. Its restoration inhibits proliferation, induces apoptosis, and suppresses tumor growth. Because NRAS also activates PI3K, miR-203 concurrently attenuates both pathways, highlighting its dual therapeutic value [72]. miR-6838-5p targets chromobox 4 (CBX4), a driver of proliferation, self-renewal, and metastasis in liver cancer stem-cells. Its loss allows CBX4-driven ERK activation, whereas its restoration suppresses HCC growth and stemness [73]. Table 4 summarizes the role of dysregulated miRNAs in the RAS/MAPK pathway.

## 7. p53 Pathway

The p53 pathway is a central tumor suppressor network that regulates cellular responses to oncogenic stress, DNA damage, and metabolic imbalance [77]. p53 functions as a tetrameric transcription factor, binding specific DNA response elements and activating targets such as p21 (cell-cycle arrest), p53 upregulated modulator of apoptosis (PUMA; apoptosis), and miR-34a (senescence). p53 also controls miRNA transcription, while in turn, miRNAs regulate p53 directly or indirectly, forming a complex feedback network [15]. p53 transactivates the miR-34 family, which suppresses nicotinamide adenine dinucleotide-dependent protein deacetylase sirtuin-1 (SIRT1) and B-cell lymphoma 2 (BCL2) to promote apoptosis and senescence. Conversely, p53 itself is modulated by miRNAs that target either its mRNA or its regulators. For example, MDM2 (murine double minute 2), the main negative regulator of p53, is inhibited by miR-192 and miR-194. Furthermore, p53 participates in miRNA maturation by interacting with the DROSHA complex after DNA damage, thereby enhancing tumor suppressor miRNA processing. Mutant p53 disrupts this mechanism, contributing to aberrant miRNA profiles in cancer [78]. Viral factors (hepatitis B virus [HBV], hepatitis C virus [HCV]) further modulate this axis, emphasizing the role of p53–miRNA crosstalk in HCC [79].

Oncogenic miRNAs: miR-146a and miR-221 are elevated in exosomes from HCC patients and correlate with progression and poor survival, supporting their role as non-invasive biomarkers [80]. miR-483-3p acts as an oncogenic partner in glucose-dependent regulation. High glucose increases miR-483-3p via hypoxia-inducible factor 1 alpha subunit (HIF1α), which suppresses PUMA and blocks apoptosis. Its inhibition restores sensitivity to apoptosis [81]. miR-519d is overexpressed due to chromosome 19 microRNA cluster (C19MC) hypomethylation. It directly targets CDKN1A/p21 and PTEN, while activating oncogenic effectors including Akt3 and tissue inhibitor of metalloproteinase 2 (TIMP2), driving growth, invasion, and therapy resistance [82]. miR-1228 binds the 3′UTR of p53, suppressing its expression and reinforcing a feedback loop where p53 loss further elevates miR-1228. High miR-1228 expression enhances proliferation, migration, and cell-cycle progression [83].

Tumor suppressor miRNAs: miR-16 and miR-192 (exosomal) have clinical biomarker potential. Low plasma miR-16 correlates with poor outcomes, while high exosomal miR-192 predicts shorter survival [80]. miR-26a activates the p53/p21 pathway by directly targeting MDM2, leading to increased p21, p27, and p53 expression. Its overexpression induces G1/S arrest and apoptosis, while its inhibition promotes tumor progression. miR-26a thus modulates the MDM2/p53 feedback loop and represents a potential therapeutic target [84]. miR-30e is downregulated in HCC and linked to enhanced cell survival. It modulates sirtuin-7 and other regulators of the p53 pathway, suggesting its therapeutic value in restoring apoptosis [79]. miR-125a-5p targets TP53-regulated inhibitor of apoptosis 1 (TRIAP1) and BCL2-like protein 2 (BCL2L2), suppressing proliferation and migration while promoting apoptosis. Its upregulation increases caspase-9 and apoptotic protease activating factor 1 (APAF1) levels, reinforcing its pro-apoptotic role [85]. miR-145-5p functions as a tumor suppressor in wild-type TP53 (tumor protein p53) tumors. Under low-glucose conditions, AMP-activated protein kinase (AMPK) activation enhances miR-145-5p while suppressing HIF1α-dependent miR-483-3p. This reciprocal regulation controls apoptosis via BCL2-binding component 3 (BBC3)/PUMA. Loss of miR-145-5p or gain of miR-483-3p confers resistance to apoptosis, highlighting their therapeutic relevance [81].

The p53–miRNA network integrates stress responses, apoptosis, and cell-cycle control. Tumor-suppressive miRNAs (e.g., miR-26a, miR-125a-5p, miR-145-5p) restore or amplify p53 activity, whereas oncogenic miRNAs (e.g., miR-483-3p, miR-519d, miR-1228) suppress it, leading to unchecked growth and resistance. Emerging therapeutic strategies include (1) delivery of tumor suppressor miRNA mimics (e.g., miR-26a, miR-125a-5p) via nanocarriers, and (2) combining miRNA modulators with MDM2 inhibitors (e.g., nutlin-(3) for synergistic p53 activation. Figure 3 depicts the interplay between tumor suppressor and oncogenic miRNAs in the p53 pathway, and Table 5 summarizes key miRNAs regulating this axis in HCC.

## 8. miRNA-Based Therapeutics in HCC: Clinical Progress and Delivery Strategies

miRNAs are emerging as important therapeutic agents in HCC, regulating oncogenic pathways and serving as biomarkers. Dysregulated miRNA expression is closely linked to hepatocarcinogenesis, influencing proliferation, invasion, apoptosis, and metastasis [89]. Accordingly, therapeutic strategies have focused on restoring tumor-suppressive miRNAs or inhibiting oncomiRs to block HCC progression [90].

The best-studied candidate is miR-34a, a p53-regulated tumor suppressor miRNA. miR-34a modulates cell-cycle arrest, apoptosis, and the DNA damage response, but is frequently downregulated in cancers due to p53 mutations or deletions [91]. To restore its function, the liposomal nanoparticle-encapsulated mimic MRX34 (liposomal miR-34a mimic) was tested in a first-in-human Phase I trial (NCT01829971) across multiple cancer types, including HCC [92]. Although MRX34 downregulated oncogenes and immune escape-related genes, the trial was terminated early due to severe immune-mediated toxicities, including fatal adverse events [93]. Despite this, MRX34 demonstrated proof-of-concept for miRNA therapy in cancer. Another candidate is the miR-193 family, which regulates apoptosis via caspase-3/7 activation and suppresses tumor development in HCC and other cancers. INT-1B3, a lipid nanoparticle-formulated miR-193a-3p mimic, is under Phase I evaluation (NCT04675996). Preclinical studies showed that 1B3 treatment upregulated PTEN and downregulated oncogenic pathways across multiple cancer cell lines, including HCC [94]. These trials illustrate both the promise and challenges of synthetic miRNA mimics.

Efficient and tumor-specific delivery remains a major hurdle in miRNA therapy. LNPs are currently the leading non-viral carriers, offering low immunogenicity, high encapsulation efficiency, and targeted delivery to the liver [95]. Gold nanoparticles (AuNPs) were successfully used to deliver miR-326 in HCC models, suppressing the 3-phosphoinositide-dependent protein kinase 1 (PDK1)/Akt/c-Myc axis and reducing EMT and tumor growth in vivo [96]. Viral vectors, including lentiviruses and adenoviruses, provide high efficiency and sustained miRNA expression but carry risks of immune activation and limited cargo capacity. In contrast, non-viral systems such as liposomes, polymers, and synthetic nanoparticles offer safer alternatives, albeit with shorter persistence [97]. Exosomes represent a novel delivery platform; as natural vesicles, they can encapsulate miRNAs and selectively deliver them to HCC cells through ligand–receptor interactions. Engineered exosomes carrying tumor-suppressive miRNAs combine high specificity with low off-target toxicity, making them attractive candidates for clinical translation.

## 9. miRNA-Based Combination Therapies

Drug resistance remains a major challenge in HCC management. Combining therapeutic miRNAs with conventional treatments has emerged as a strategy to enhance efficacy, overcome resistance, and reduce drug toxicity by allowing lower dosages [98]. Raniolo et al. designed DNA nanocages (Fol-miR21-NCs) that trap oncogenic miR-21 while co-delivering doxorubicin. Blocking miR-21 restored tumor suppressors such as PTEN and programmed cell death 4 (PDCD4), reduced tumor growth, and enhanced doxorubicin sensitivity [99]. Similarly, overexpression of miR-34a increased the efficacy of doxorubicin in HepG2 cells. Combination treatment induced G1 arrest and reduced drug-resistance proteins (multidrug resistance protein 1 [MDR1]/P-glycoprotein and AXL receptor tyrosine kinase [AXL]), likely through p53 regulation [100]. Hassan et al. combined a miR-122 mimic with a miR-221 inhibitor in mice, which reduced inflammation, proliferation, and angiogenesis more effectively than either treatment alone. This dual approach downregulated cyclin D1, TGF-β, and β-catenin, improved liver architecture, and targeted SUMO-specific protease 1 (SENP1) and ADP-ribosylation factor 4 (ARF4) to induce apoptosis [101]. miR-338-3p is often downregulated in HCC tissues. Its restoration suppressed HIF-1α, inhibited growth, and increased sorafenib sensitivity in cell and animal models [102]. Wang et al. showed that miR-145-5p suppresses histone deacetylase 11 (HDAC11), which is upregulated in sorafenib-resistant HCC. Loss of miR-145-5p promoted resistance and metastasis, whereas restoring it reduced proliferation and drug resistance by inhibiting cytochrome P450-mediated metabolism [103]. miR-3622b-5p promotes apoptosis and sensitizes HCC cells to cisplatin, supporting its use in combination therapy [104]. Gold nanomaterials can release miRNA inhibitors, mimics, or conjugates, restoring miRNA function and reversing resistance [105]. Externally repurposed drugs also offer new approaches. For example, amiodarone, a widely used anti-arrhythmic, suppresses HCC by inducing autophagy-mediated degradation of oncogenic miR-224 [106].

Combination therapies that integrate miRNA mimics or inhibitors with chemotherapeutics (cisplatin, doxorubicin, sorafenib) or repurposed drugs (amiodarone) show superior efficacy compared with monotherapies. Advances in delivery systems such as DNA nanocages and AuNPs further strengthen their translational potential, providing new avenues for personalized and more effective HCC treatment.

## 10. Clinical Trials on miRNA Biomarkers and Therapies in HCC

Preclinical and clinical studies are increasingly investigating miRNAs as biomarkers for diagnosis, prognosis, and treatment response in HCC. These trials explore circulating miRNAs, exosomal signatures, and isomiRs, aiming to improve patient selection and treatment efficacy. NCT05148572 validates circulating miRNA signatures in patients undergoing surgical resection to predict recurrence risk. NCT04965259 (AHCC10 ELEGANCE study) develops a miRNA-based diagnostic kit for early HCC detection in high-risk populations; it also integrates magnetic resonance imaging-based artificial intelligence (AI) algorithms and microbiome/metabolome profiling to predict disease progression. NCT04720430 evaluates circulating miRNAs as predictors of response to loco-regional therapy (LRT) before liver transplantation, aiming to refine treatment decisions and patient selection. NCT05449847 investigates plasma miR-21 and miR-126 as biomarkers for distinguishing uncomplicated vs. complicated HCV cases, including HCC, cirrhosis, and steatohepatitis. NCT02412579 explores genetic markers and circulating miRNA isoforms (isomiRs) as prognostic tools in liver transplantation candidates. NCT06342414 develops an exosome-based liquid biopsy combined with machine learning to differentiate HCC from intrahepatic cholangiocarcinoma, providing a cost-effective presurgical diagnostic tool.

Collectively, these trials highlight the potential of circulating and exosomal miRNAs, combined with advanced technologies such as AI and metabolomic profiling, to serve as non-invasive biomarkers for diagnosis, recurrence monitoring, prognosis, and treatment response in HCC. Table 6 summarizes key ongoing clinical trials evaluating circulating miRNAs and isomiRs as biomarkers in HCC.

## 11. Challenges and Future Perspectives

Despite progress in cancer research, no existing therapy reliably eliminates all cancer cells or consistently improves long-term survival rates in HCC. Given the strong link between miRNAs and tumor biology, miRNA-based strategies offer great promise. However, several challenges must be addressed before clinical translation. A single miRNA may regulate multiple mRNAs, some with opposing biological effects, leading to unexpected toxicities such as neurotoxicity and immunotoxicity. Comprehensive mapping of miRNA–mRNA interactions and rigorous evaluation of off-target effects are essential to minimize adverse outcomes. Moreover, targeting one gene alone may not be sufficient for therapeutic efficacy, emphasizing the need to better understand the broader signaling networks regulated by miRNAs.

miRNAs have short half-lives in vivo and are prone to rapid degradation. Effective delivery systems, such as LNPs, dendrimers, or engineered exosomes, are required to protect miRNAs, ensure their stability, and direct them to tumor tissues. Even with optimized carriers, immune recognition of miRNA therapeutics remains a concern, necessitating chemical modifications to reduce immunogenicity. Before clinical application, extensive preclinical testing is needed to confirm not only therapeutic efficacy but also long-term safety. Personalized approaches will likely be crucial, as genetic heterogeneity among HCC patients may determine which miRNA therapies are most effective. Tailoring treatment to molecular and genetic profiles could maximize efficacy while minimizing side effects. By advancing our understanding of the complex miRNA–mRNA regulatory networks, improving delivery technologies, and conducting well-designed clinical trials, miRNA-based therapies may evolve into powerful tools for early diagnosis, prognosis, and treatment of HCC.

## 12. Conclusions

HCC remains a global health challenge, with limited treatment options and poor survival outcomes in advanced stages. Dysregulated miRNAs drive hepatocarcinogenesis by modulating tumor suppressors and oncogenes across key oncogenic pathways, including PI3K/Akt/mTOR, Hippo–YAP/TAZ, Wnt/β-catenin, RAS/MAPK, and p53. These regulatory interactions promote tumor progression, metastasis, and therapy resistance. Among these networks, the PI3K/Akt/mTOR axis appears especially promising for therapeutic intervention, as it integrates signals from multiple HCC pathways that regulate proliferation, survival, and drug resistance. By targeting multiple oncogenes and signaling proteins simultaneously, miRNAs represent both potential diagnostic biomarkers and therapeutic tools. Recent advances in nanoparticle-based delivery and combination strategies underscore their translational potential. As biomarkers, circulating miRNAs have demonstrated high accuracy, with 85–90% specificity for early HCC detection, supporting their use in screening and prognosis. Looking ahead, miRNA-based therapies may revolutionize HCC management by enabling multi-targeted, pathway-specific interventions. Continued research is essential to refine delivery systems, minimize off-target effects, and establish clinical feasibility, ultimately paving the way for personalized and more effective treatments.

## Figures and Tables

**Figure 1 ijms-26-08365-f001:**
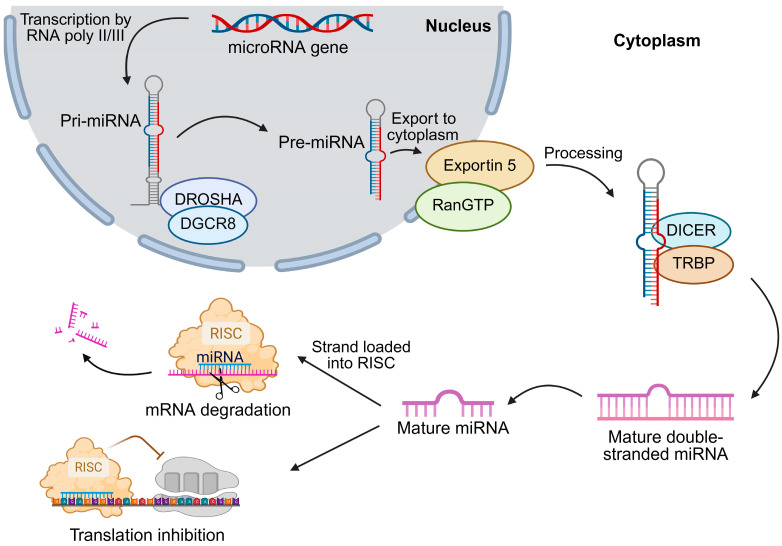
Biogenesis of microRNAs. RNA polymerase II typically transcribes miRNA genes to produce primary transcripts (pri-miRNAs). The microprocessor complex—comprising DROSHA and the RNA-binding protein DGCR8—cleaves pri-miRNAs to generate ~85-nt precursor miRNAs (pre-miRNAs). After nuclear export via the RanGTP/Exportin-5 complex, DICER processes pre-miRNAs into ~20–22-nt miRNA/miRNA duplexes. The mature strand is then incorporated into the RNA-induced silencing complex (RISC), which represses translation or induces mRNA degradation, depending on the degree of base-pairing complementarity. DGCR8—DiGeorge syndrome critical region 8; DICER—RNase III endoribonuclease; DROSHA—RNase III endoribonuclease; miRNA—microRNA; pre-miRNA—precursor microRNA; pri-miRNA—primary microRNA; RanGTP—RAS-related nuclear protein; RISC—RNA-induced silencing complex; TRBP—TAR RNA-binding protein.

**Figure 2 ijms-26-08365-f002:**
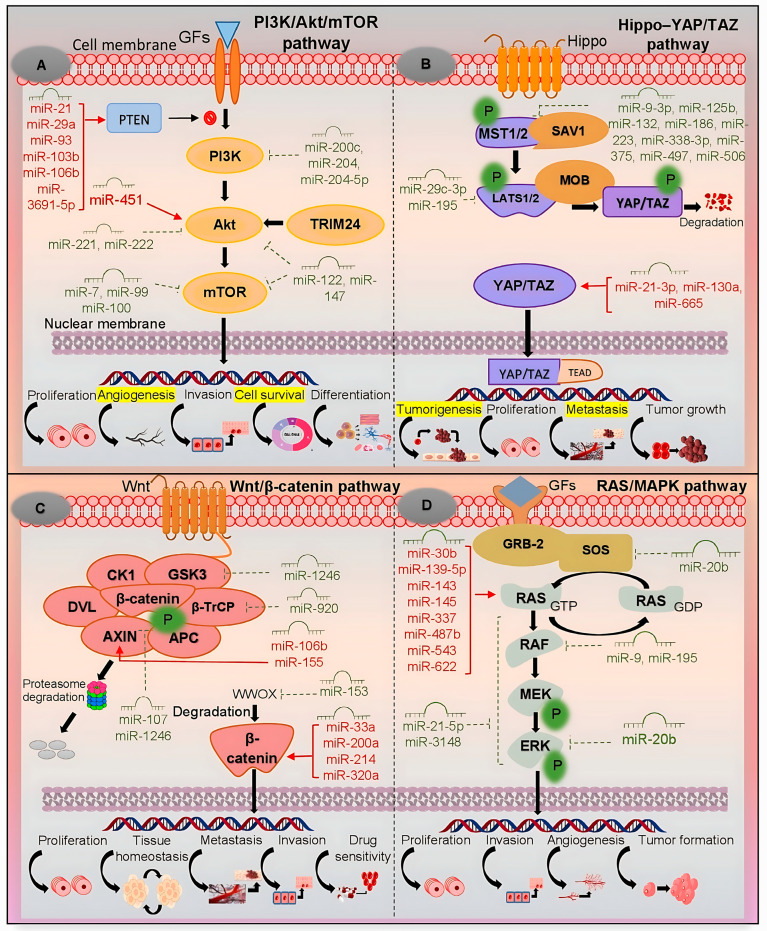
Signaling pathways in cancer progression include the PI3K/Akt/mTOR, Hippo–YAP/TAZ, Wnt/β-catenin, and RAS/MAPK pathways. The figure highlights the regulatory roles of miRNAs in these networks, either promoting or inhibiting tumor-related processes such as proliferation, invasion, angiogenesis, metastasis, apoptosis, and drug resistance. (**A**) In the PI3K/Akt/mTOR pathway, miRNAs regulate central effectors including PI3K, Akt, and mTOR, thereby influencing cell survival and differentiation. (**B**) In the Hippo–YAP/TAZ pathway, miRNAs control upstream kinases and YAP/TAZ transcriptional activity, shaping tumorigenesis and metastatic potential. (**C**) In the Wnt/β-catenin pathway, miRNAs modulate β-catenin stability and transcriptional activity, contributing to tissue homeostasis, invasion, and therapy sensitivity. (**D**) In the RAS/MAPK pathway, miRNAs target RAS, RAF, MEK, and ERK signaling, driving tumor growth, angiogenesis, and invasive behavior. miRNAs shown in green function as tumor suppressors, whereas those in red act as oncomiRs, underscoring their dual roles in cancer. Akt—protein kinase B; APC—adenomatous polyposis coli; AXIN—axis inhibition protein; β-TrCP—beta-transducin repeat-containing protein; CK1—casein kinase 1; DVL—Dishevelled segment polarity protein; ERK—extracellular signal-regulated kinase; GDP—guanosine diphosphate; GF—growth factor; GRB-2—growth factor receptor-bound protein 2; GSK3—glycogen synthase kinase 3; GTP—guanosine triphosphate; LATS1/2—large tumor suppressor kinases 1 and 2; MAPK—mitogen-activated protein kinase; MEK—MAP/ERK kinase; miR—microRNA; MOB—Mps one binder; MST1/2—mammalian Ste20-like serine/threonine kinases 1 and 2; mTOR—mechanistic (mammalian) target of rapamycin; P—protein; PI3K—phosphatidylinositol 3-kinase; PTEN—phosphatase and tensin homolog; RAF—rapidly accelerated fibrosarcoma; RAS—rat sarcoma small GTPase; SAV1—Salvador family WW domain-containing protein 1; SOS—Son of Sevenless; TEAD—TEA domain transcription factor; TRIM24—tripartite motif-containing 24; Wnt—Wingless/Int-1; WWOX—WW domain-containing oxidoreductase; YAP/TAZ—Yes-associated protein/transcriptional co-activator with PDZ-binding motif.

**Figure 3 ijms-26-08365-f003:**
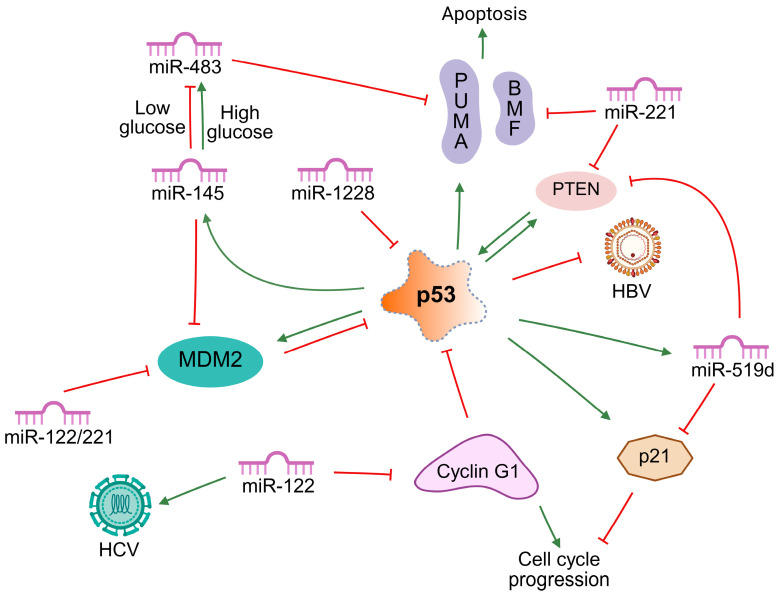
Interactions between key miRNAs (miR-122, miR-145, miR-221, miR-483-3p, miR-519d, miR-1228) and the p53 pathway, including its regulators (MDM2, PTEN, cyclin G1). The figure illustrates their roles in apoptosis, cell-cycle control, and viral modulation (HBV/HCV), showing how miRNAs either suppress or enhance p53 activity. Green arrows indicate gene induction or protein activation, whereas red lines denote gene repression or protein inhibition. BMF—B-cell lymphoma 2 modifying factor; HBV—hepatitis B virus; HCV—hepatitis C virus; MDM2—murine double minute 2; miR—microRNA; p21—cyclin-dependent kinase inhibitor 1A; p53—tumor protein p53; PTEN—phosphatase and tensin homolog; PUMA—p53 upregulated modulator of apoptosis.

**Table 1 ijms-26-08365-t001:** Differentially expressed miRNAs that regulate the PI3K/Akt/mTOR pathway in HCC.

miRNA (↑/↓)	Target(s)	Biochemical Role (Gene/Protein)	Affected Biological Process(es)	miRNA Function	Ref.
PI3K/Akt/mTOR Pathway					
miR-92a-3p ↑	N-cadherin; vimentin	N-cadherin: adhesion protein; vimentin: cytoskeletal EMT marker	Proliferation; migration; invasion; EMT; apoptosis	OncomiR	[31]
miR-93 ↑	PTEN; CDKN1A	PTEN: PIP3 phosphatase; CDKN1A (p21): cell-cycle inhibitor	Tumor progression; migration; invasion; apoptosis	OncomiR	[24]
miR-106b-5p ↑	FOG2	Transcriptional cofactor; regulates stem-cell differentiation	Proliferation; stem-cell differentiation and self-renewal	OncomiR	[32]
miR-660-5p ↑	KLF3; YWHAH	KLF3: transcription factor; YWHAH (14-3-3η): scaffold protein	Proliferation; migration; invasion; EMT	OncomiR	[33]
miR-3691-5p ↑	PTEN	PIP3 phosphatase; inhibits PI3K/Akt signaling	Tumor progression; cell proliferation	OncomiR	[27]
miR-30b-3p ↓	TRIM27	E3 ubiquitin ligase; activates PI3K/Akt signaling	Proliferation; migration; invasion; EMT	Tumor suppressor	[28]
miR-34a ↓	HDAC1; MACC1	HDAC1: histone deacetylase; MACC1: activates c-Met/PI3K/Akt signaling	Cell proliferation; apoptosis	Tumor suppressor	[34]
miR-125a-5p ↓	MACC1	Activates c-Met/PI3K/Akt signaling; promotes metastasis	Proliferation; metastasis; apoptosis	Tumor suppressor	[23]
miR-133b ↓	EGFR	Receptor tyrosine kinase; activates MAPK/PI3K pathways	Cell proliferation; apoptosis	Tumor suppressor	[29]
miR-148a ↓	DR5	Death receptor; mediates caspase-dependent apoptosis	S phase regulation; apoptosis	Tumor suppressor	[35]
miR-199a/b-5p ↓	ROCK1	Rho kinase; regulates cytoskeleton and motility	Proliferation; migration; invasion	Tumor suppressor	[36]
miR-1914 ↓	GPR39	Zinc-activated GPCR; suppresses PI3K/Akt/mTOR signaling	Cell-cycle control; apoptosis; tumor growth	Tumor suppressor	[30]

↑ indicates upregulation of the respective miRNA, while ↓ indicates its downregulation in the table. CDKN1A—cyclin-dependent kinase inhibitor 1A; DR5—death receptor 5; EGFR—epidermal growth factor receptor; EMT—epithelial–mesenchymal transition; FOG2—friend of GATA 2; GPCR—G protein-coupled receptor; GPR39—G protein-coupled receptor 39; HCC—hepatocellular carcinoma; HDAC1—histone deacetylase 1; KLF3—Krüppel-like factor 3; MACC1—metastasis-associated in colon cancer 1; MAPK—mitogen-activated protein kinase; miRNA—microRNA; N-cadherin—neural cadherin; PI3K/Akt/mTOR—phosphatidylinositol 3-kinase/protein kinase B/mammalian target of rapamycin; PIP3—phosphatidylinositol 3,4,5-trisphosphate; PTEN—phosphatase and tensin homolog; ROCK1—Rho-associated coiled-coil-containing protein kinase 1; TRIM27—tripartite motif-containing 27; vimentin—type III intermediate filament protein; YWHAH—tyrosine 3-monooxygenase/tryptophan 5-monooxygenase activation protein, eta.

**Table 2 ijms-26-08365-t002:** Differentially expressed miRNAs regulating the Hippo–YAP/TAZ pathway in HCC.

miRNA (↑/↓)	Target(s)	Biochemical Role (Gene/Protein)	Affected Biological Process(es)	miRNA Function	Ref.
Hippo–YAP/TAZ Pathway					
miR-21 ↑	YOD1	Deubiquitinase; stabilizes YAP/TAZ by preventing degradation	Proliferation	OncomiR	[38]
miR-21-3p ↑	SMAD7	Negative regulator of TGF-β signaling; loss leads to YAP1 activation	Tumor progression	OncomiR	[39]
miR-130a ↑	VGLL4	TEAD-binding protein; competes with YAP/TAZ to inhibit Hippo signaling	Over-proliferation; tumorigenesis	OncomiR	[46]
miR-135b ↑	MST1	Serine/threonine kinase, phosphorylates LATS1/2 to activate Hippo pathway	Proliferation; invasion; migration	OncomiR	[40]
miR-512-3p ↑	LATS2	Kinase; phosphorylates/inactivates YAP/TAZ	Proliferation; invasion; migration	OncomiR	[41]
miR-1307-3p ↑	LATS1	Kinase in Hippo pathway; phosphorylates YAP/TAZ	Proliferation; migration; invasion	OncomiR	[42]
miR-9-3p ↓	TAZ	Transcriptional co-activator; binds TEADs to promote proliferation	Proliferation; invasion; migration	Tumor suppressor	[46]
miR-29c-3p ↓	DNMT3B	DNA methyltransferase; epigenetically silences tumor suppressors	Proliferation; migration; tumor growth	Tumor suppressor	[44]
miR-186 ↓	YAP1	Transcriptional co-activator; drives oncogenic gene expression	Tumorigenesis	Tumor suppressor	[38]
miR-375 ↓	YAP1; CTGF; TEAD4	YAP1: co-activator; CTGF: pro-fibrotic factor; TEAD4: transcription factor	Proliferation; tumor growth	Tumor suppressor	[45]

↑ indicates upregulation of the respective miRNA, while ↓ indicates its downregulation in the table. CTGF—connective tissue growth factor; DNMT3B—DNA methyltransferase 3B; LATS1/2—large tumor suppressor kinases 1 and 2; miRNA—microRNA; MST1—mammalian sterile 20-like kinase 1; SMAD7—mothers against decapentaplegic homolog 7; TAZ—transcriptional co-activator with PDZ-binding motif; TEAD4—TEA domain transcription factor 4; TGF-β—transforming growth factor beta; VGLL4—vestigial-like family member 4; YAP1—Yes-associated protein 1; YOD1—YOD1 deubiquitinase.

**Table 3 ijms-26-08365-t003:** Differentially expressed miRNAs regulating the Wnt/β-catenin pathway in HCC.

miRNA (↑/↓)	Target(s)	Biochemical Role (Gene/Protein)	Affected Biological Process(es)	miRNA Function	Ref.
Wnt/β-Catenin Pathway					
miR-19a-3p, miR-376c-3p ↑	SOX6	Binds β-catenin; blocks nuclear translocation	Proliferation; invasion; migration	OncomiR	[49]
miR-197 ↑	AXIN2; DKK2; NKD1	Negative Wnt regulators	Invasion; metastasis	OncomiR	[50]
miR-409-3p ↑	BRF2	Transcription factor; activates Wnt/β-catenin	Invasion; metastasis	OncomiR	[58]
miR-452-5p ↑	CDKN1B	Cell-cycle inhibitor; loss promotes stemness	Cancer stem-cell differentiation; self-renewal	OncomiR	[60]
miR-500 ↑	SFPR2; GSK3β	Wnt antagonists; inhibition activates β-catenin	Invasion; migration; prognosis	OncomiR	[51]
miR-1246 ↑	RORα; CADM1; GSK3β; AXIN2	AXIN2/GSK3β: degrade β-catenin; RORα: nuclear receptor modulating Wnt	Proliferation; invasion; migration; drug resistance	OncomiR	[61]
miR-194 ↓	PRC1; β-catenin	Cytokinesis regulator; stabilizes β-catenin	Proliferation; invasion; migration	Tumor suppressor	[53]
miR-329-3p ↓	USP22	Deubiquitinase; stabilizes β-catenin	Proliferation; migration	Tumor suppressor	[54]
miR-342 ↓	CXCL12	Chemokine-activating Wnt/β-catenin	Proliferation; apoptosis	Tumor suppressor	[62]
miR-361-5p ↓	WT1	Transcriptional regulator; modulates EMT	Proliferation; invasion; migration	Tumor suppressor	[56]
miR-485-5p ↓	WBP2	Scaffold protein; promotes β-catenin nuclear entry	Proliferation; invasion; migration	Tumor suppressor	[59]

↑ indicates upregulation of the respective miRNA, while ↓ indicates its downregulation in the table. AXIN2—axis inhibition protein 2; β-catenin—transcriptional co-activator in Wnt signaling; BRF2—transcription factor IIIB, 50 kDa subunit; CADM1—cell adhesion molecule 1; CDKN1B—cyclin-dependent kinase inhibitor 1B; CXCL12—CXC motif chemokine 12; DKK2—Dickkopf-related protein 2; EMT—epithelial–mesenchymal transition; GSK3β—glycogen synthase kinase 3 beta; HCC—hepatocellular carcinoma; miR—microRNA; NKD1—naked cuticle homolog 1; PRC1—protein regulator of cytokinesis 1; RORα—retinoic acid-related orphan receptor alpha; SFPR2—secreted frizzled-related protein 2; SOX6—SRY-box transcription factor 6; USP22—ubiquitin-specific peptidase 22; WBP2—WW domain-binding protein 2; WT1—Wilms tumor 1.

**Table 4 ijms-26-08365-t004:** Differentially expressed miRNAs regulating the RAS/MAPK pathway in HCC.

miRNA (↑/↓)	Target(s)	Biochemical Role (Gene/Protein)	Affected Biological Process(es)	miRNA Function	Ref.
RAS/MAPK Pathway					
miR-29 ↑	TTP	RNA-binding protein; destabilizes mRNAs	Invasion; metastasis; angiogenesis	OncomiR	[74]
miR-107 ↑	CPEB3	RNA-binding protein; regulates mRNA translation	Proliferation; invasion; migration	OncomiR	[75]
miR-126-3p ↑	SPRED1	Negative regulator of RAS/RAF/MAPK signaling	Invasion; metastasis; recurrence; sorafenib resistance	OncomiR	[66]
miR-148a-3p ↑	MTF1	Transcription factor; MAPK effector in AP-1 complex	Proliferation; metastasis; apoptosis	OncomiR	[71]
miR-330-5p ↑	SPRY2	Negative regulator of RTK/MAPK signaling	Proliferation; tumor growth	OncomiR	[67]
miR-30a ↓	Beclin 1; ATG5	Autophagy regulators; promote cell survival	Tumor growth; apoptosis	Tumor suppressor	[68]
miR-101 ↓	EZH2	Histone methyltransferase; epigenetically silences tumor suppressors	Proliferation; metastasis	Tumor suppressor	[69]
miR-122 ↓	IGF-1R	Tyrosine kinase receptor; activates RAS/MAPK and PI3K/Akt	Proliferation; invasion; migration; survival	Tumor suppressor	[70]
miR-203 ↓	NRAS	GTPase; activates RAS/MAPK and PI3K/Akt	Proliferation; apoptosis; tumor growth	Tumor suppressor	[72]
miR-296-5p ↓	NRG1	Growth factor; activates ERBB receptors upstream of RAS	Invasion; migration; EMT	Tumor suppressor	[76]
miR-6838-5p ↓	CBX4	Chromobox protein; promotes ERK signaling	CSC self-renewal; metastasis	Tumor suppressor	[73]

↑ indicates upregulation of the respective miRNA, while ↓ indicates its downregulation in the table. Akt—protein kinase B; AP-1—activator protein-1; ATG5—autophagy protein 5; Beclin 1—autophagy-related protein Beclin 1; CBX4—chromobox 4; CPEB3—cytoplasmic polyadenylation element-binding protein 3; CSC—cancer stem cell; EMT—epithelial–mesenchymal transition; ERBB—Erb-B2 receptor tyrosine kinase family; ERK—extracellular signal-regulated kinase; EZH2—enhancer of zeste homolog 2; IGF-1R—insulin-like growth factor 1 receptor; MAPK—mitogen-activated protein kinase; miR—microRNA; mRNA—messenger RNA; MTF1—metal regulatory transcription factor 1; NRAS—neuroblastoma RAS viral oncogene homolog; NRG1—neuregulin 1; PI3K—phosphatidylinositol 3-kinase; PTEN—phosphatase and tensin homolog; RAF—rapidly accelerated fibrosarcoma; RAS—rat sarcoma small GTPase; RTK—receptor tyrosine kinase; SPRED1—sprouty-related EVH1 domain-containing protein 1; SPRY2—sprouty RTK (receptor tyrosine kinase) signaling antagonist 2; TTP—tristetraprolin.

**Table 5 ijms-26-08365-t005:** Differentially expressed miRNAs regulating the p53 pathway in HCC.

miRNA (↑/↓)	Target(s)	Biochemical Role (Gene/Protein)	Affected Biological Process(es)	miRNA Function	Ref.
p53 Pathway					
miR-221 ↑	CDKN2B; CDKN2C; PTEN; TIMP3; MDM2	CDKN2B/C: cell-cycle inhibitors; PTEN: PI3K/Akt antagonist; TIMP3: metalloproteinase inhibitor	Proliferation; tumor growth	OncomiR	[80]
miR-483-3p ↑	BBC3/PUMA; IGF2	PUMA: pro-apoptotic protein; IGF2: growth factor	Cell survival; tumor growth	OncomiR	[81]
miR-519d ↑	CDKN1A/p21; PTEN; MDM2; TIMP2	p21: CDK inhibitor; TIMP2: metalloproteinase inhibitor	Proliferation; invasion; apoptosis	OncomiR	[82]
miR-1228 ↑	p53	Master tumor suppressor; regulates cell-cycle and apoptosis	Cell-cycle progression; migration	OncomiR	[83]
miR-23a ↓	XIAP	Inhibitor of apoptosis; blocks caspase activation	Tumor development; progression	Tumor suppressor	[86]
miR-26a ↓	MDM2	E3 ubiquitin ligase; degrades p53	Cell-cycle arrest (p21/p27); apoptosis	Tumor suppressor	[84]
miR-30e ↓	SIRT7	NAD^+^-dependent deacetylase; stabilizes oncoproteins	Tumor growth; progression	Tumor suppressor	[79]
miR-34 ↓	c-Met; caspase-3; ERK1/2	c-Met: RTK activating survival; caspase-3: apoptosis executor; ERK1/2: MAPK kinases	Tumor progression; invasion; metastasis	Tumor suppressor	[87]
miR-122 ↓	MDM2; cyclin G1	MDM2: E3 ubiquitin ligase for p53; cyclin G1: cell-cycle regulator	Tumor growth; invasion; metastasis	Tumor suppressor	[79]
miR-125a-5p, miR-125b ↓	LATS1; TRIAP1; BCL2L2	LATS1: Hippo kinase; TRIAP1/BCL2L2: anti-apoptotic proteins	Proliferation; invasion; apoptosis	Tumor suppressor	[88]
miR-145-5p ↓	MDM2	Negative regulator of p53 stability	Tumor progression; metastasis	Tumor suppressor	[81]

↑ indicates upregulation of the respective miRNA, while ↓ indicates its downregulation in the table. Akt—protein kinase B; BBC3—B-cell lymphoma 2-binding component 3; BCL2L2—B-cell lymphoma 2-like protein 2; CDK—cyclin-dependent kinase; CDKN1A—cyclin-dependent kinase inhibitor 1A; CDKN2B—cyclin-dependent kinase inhibitor 2B; CDKN2C—cyclin-dependent kinase inhibitor 2C; c-Met—hepatocyte growth factor receptor; ERK1/2—extracellular signal-regulated kinases 1 and 2; HCC—hepatocellular carcinoma; IGF2—insulin-like growth factor 2; LATS1—large tumor suppressor kinase 1; MAPK—mitogen-activated protein kinase; MDM2—murine double minute 2; miR—microRNA; p21—tumor protein p21; p27—tumor protein p27; p53—tumor protein p53; PI3K—phosphatidylinositol 3-kinase; PTEN—phosphatase and tensin homolog; PUMA—p53 upregulated modulator of apoptosis; RTK—receptor tyrosine kinase; SIRT7—nicotinamide adenine dinucleotide-dependent protein deacetylase sirtuin-7; TIMP2—tissue inhibitor of metalloproteinase 2; TIMP3—tissue inhibitor of metalloproteinase 3; TRIAP1—TP53-regulated inhibitor of apoptosis 1; XIAP—X-linked inhibitor of apoptosis protein.

**Table 6 ijms-26-08365-t006:** Ongoing and completed clinical trials investigating miRNA biomarkers in HCC.

Trial ID	Treatment	Disease	Patients (n)	Study Aim	Status
NCT02412579	Observational	HCC	40	IsomiR profiling as a non-invasive biomarker in liver transplantation candidates	Completed
NCT04720430	LRT	HCC	7	Circulating miRNAs to predict LRT response before transplantation	Completed
NCT04965259	N/A	HCC	2000	Development of miRNA-based diagnostic kit for early detection in high-risk patients (with MRI-AI, microbiome/metabolome profiling)	Ongoing
NCT05148572	Surgical resection	HCC	100	Validation of circulating miRNAs for recurrence prediction	Ongoing
NCT05449847	Observational	HCV (±HCC, cirrhosis, steatohepatitis)	100	Plasma miR-21 and miR-126 as biomarkers to predict HCC in HCV patients	Completed
NCT06342414	N/A	HCC/intrahepatic cholangiocarcinoma	400	Exosomal miRNAs and machine learning for presurgical diagnosis and differentiation of liver cancers	Ongoing

AI—artificial intelligence; HCC—hepatocellular carcinoma; HCV—hepatitis C virus; isomiR—microRNA isoform; LRT—loco-regional therapy; miRNA—microRNA; MRI—magnetic resonance imaging; N/A—not applicable.

## Data Availability

Not applicable.

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
