# Peer review of "Dysregulation of MicroRNAs in Hepatocellular Carcinoma: Targeting Oncogenic Signaling Pathways for Innovative Therapies"

_ijms, 2025, doi:10.3390/ijms26178365_

Round 1

Reviewer 1 Report

Comments and Suggestions for Authors

In this manuscript, Zarlashat and colleagues examine the signaling pathways involved in hepatocellular carcinoma (HCC), emphasizing the role of microRNAs (miRNAs) and their potential as therapeutic targets.
HCC is a common and deadly cancer, with its development driven by the activation of oncogenes and suppression of tumor suppressor proteins. MicroRNAs (miRNAs) play a critical role in regulating gene expression and are involved in both promoting (as oncomiRs) and suppressing cancer. Their dysregulation can alter key signaling pathways, such as PI3K/AKT/mTOR, Hippo/YAP, Wnt/β-catenin, RAS/MAPK, and p53, contributing to HCC progression. miRNA-based therapies show promise in restoring tumor-suppressing miRNAs or inhibiting oncogenic ones, potentially improving treatment outcomes. The manuscript highlights the therapeutic and diagnostic potential of miRNAs in HCC, as well as current clinical trials exploring their use for early detection, recurrence prediction, and non-invasive cancer differentiation.
Overall, the review is well-written and clearly presented. However, it largely reiterates existing literature and does not offer significant new insights into the topic. Despite the title suggesting a focus on innovative therapeutic strategies for treating HCC, the manuscript primarily reads as a broad overview of signalling pathways involved in HCC pathogenesis, more akin to a textbook summary than a focused review on novel therapies.
The most compelling and potentially valuable section, discussing therapeutic strategies, is limited to a brief paragraph (paragraph 8), which feels insufficient given the paper’s stated aim. Notably, the clinical trials cited by the authors mainly focus on the use of miRNAs as diagnostic and prognostic biomarkers, rather than as therapeutic agents.
To strengthen the manuscript and make it more suitable for publication, the authors should consider expanding this section significantly. A deeper exploration of studies investigating the therapeutic potential of miRNAs, including delivery methods, preclinical/clinical outcomes, and associated challenges, would make the review more impactful and aligned with its title.
Minor comments: The abstract is overly detailed, particularly with the listing of multiple signalling pathways. It would be more effective if it provided a concise overview rather than attempting to summarise all pathways discussed in the main text.

Author Response

Reviewer 1

In this manuscript, Zarlashat and colleagues examine the signaling pathways involved in hepatocellular carcinoma (HCC), emphasizing the role of microRNAs (miRNAs) and their potential as therapeutic targets.
HCC is a common and deadly cancer, with its development driven by the activation of oncogenes and suppression of tumor suppressor proteins. MicroRNAs (miRNAs) play a critical role in regulating gene expression and are involved in both promoting (as oncomiRs) and suppressing cancer. Their dysregulation can alter key signaling pathways, such as PI3K/AKT/mTOR, Hippo/YAP, Wnt/β-catenin, RAS/MAPK, and p53, contributing to HCC progression. miRNA-based therapies show promise in restoring tumor-suppressing miRNAs or inhibiting oncogenic ones, potentially improving treatment outcomes. The manuscript highlights the therapeutic and diagnostic potential of miRNAs in HCC, as well as current clinical trials exploring their use for early detection, recurrence prediction, and non-invasive cancer differentiation.

Response: Thank you for the positive and accurate summary of our work. We are pleased that the manuscript conveys the role of miRNAs in HCC signaling pathways and their therapeutic and diagnostic potential.
Overall, the review is well-written and clearly presented. However, it largely reiterates existing literature and does not offer significant new insights into the topic. Despite the title suggesting a focus on innovative therapeutic strategies for treating HCC, the manuscript primarily reads as a broad overview of signalling pathways involved in HCC pathogenesis, more akin to a textbook summary than a focused review on novel therapies.

Response: We appreciate the reviewer's observation. In the revised version, we have included additional recent studies specifically highlighting emerging therapeutic strategies for HCC to strengthen the focus on novel therapeutic perspectives.
The most compelling and potentially valuable section, discussing therapeutic strategies, is limited to a brief paragraph (paragraph 8), which feels insufficient given the paper’s stated aim. Notably, the clinical trials cited by the authors mainly focus on the use of miRNAs as diagnostic and prognostic biomarkers, rather than as therapeutic agents.

Response: We appreciate this comment. In response, we have expanded this section by including additional data and recent studies focusing on the use of miRNAs as therapeutic agents, not just as diagnostic and prognostic biomarkers. These additions strengthen the discussion of therapeutic strategies and align the manuscript more closely with its stated aim.

To strengthen the manuscript and make it more suitable for publication, the authors should consider expanding this section significantly. A deeper exploration of studies investigating the therapeutic potential of miRNAs, including delivery methods, preclinical/clinical outcomes, and associated challenges, would make the review more impactful and aligned with its title.

Response: We appreciate the reviewer’s suggestion. We have expanded the discussion by incorporating studies on the therapeutic potential of miRNAs, including delivery approaches, preclinical and clinical outcomes, as well as related limitations. These aspects are now addressed in the sections “Developing Promising Therapeutic Strategies for miRNA in HCC Treatment” and “Challenges and Future Perspectives.”
Minor comments: The abstract is overly detailed, particularly with the listing of multiple signalling pathways. It would be more effective if it provided a concise overview rather than attempting to summarise all pathways discussed in the main text.

Response: We thank the reviewer for this valuable suggestion. We have revised the abstract accordingly, removing the detailed listing of signaling pathways and instead providing a concise overview to better reflect the scope of the review.

Reviewer 2 Report

Comments and Suggestions for Authors

This review article titled “Dysregulation of MicroRNAs in Hepatocellular Carcinoma: Targeting Oncogenic Signaling Pathways for Innovative Therapies” by Zarlashat and collegues is interesting. This review describes 6 cellular pathways that are dysregulated in hepatocellular carcinoma and the role of miRNAs responsible for this dysregulation. This review also describes how these dysregulated miRNAs can be utilized to develop innovative therapies for hepatocellular carcinoma. This review is well written, easy to read and include current information related to this field.

Author Response

Reviewer 2

This review article titled “Dysregulation of MicroRNAs in Hepatocellular Carcinoma: Targeting Oncogenic Signaling Pathways for Innovative Therapies” by Zarlashat and collegues is interesting. This review describes 6 cellular pathways that are dysregulated in hepatocellular carcinoma and the role of miRNAs responsible for this dysregulation. This review also describes how these dysregulated miRNAs can be utilized to develop innovative therapies for hepatocellular carcinoma. This review is well written, easy to read and include current information related to this field.

Response: Thank you very much for your positive feedback on our review article “Dysregulation of MicroRNAs in Hepatocellular Carcinoma: Targeting Oncogenic Signaling Pathways for Innovative Therapies.” We are glad to hear that you found the manuscript well written, informative, and easy to follow. Your encouraging remarks motivate us to continue our efforts in exploring the therapeutic potential of miRNAs in HCC and presenting updated insights for the research community.

Reviewer 3 Report

Comments and Suggestions for Authors

     This manuscript is a review of microRNAs (miRs) potentially involved in liver carcinogenesis. Such RNAs are classified into two main categories: oncomiRs downregulate the expression of specific tumor suppressor proteins and are overexpressed in tumor cells; and tumor suppressor miRs that knock down the expression of specific oncoproteins and have reduced expression in cancer cells. The authors discuss some of the miRs that affect key signaling pathways involved in the regulation of cell division, survival, migration, invasion, metastasis etc., pathways that are implicated in liver carcinogenesis. MicroRNA-regulated signaling pathways covered by the review are the PI3K-, Hippo-, Wnt-, RAS/ERK- and p53-pathways. The review appears to be based on literature search (research or clinical contribution by the authors are not known by this reviewer). The manuscript is written in clear, polished language. It discusses a field of medical science of outstanding importance.

     This reviewer, however, has several major problems with the manuscript:

  1. The audience targeted by this review are presumably medical personnel, many of them not having up-to-date knowledge in molecular biology and signal transduction. Therefore, a brief description of miR metabolism (transcription, processing etc.) and mechanism of action (perhaps with a simple figure) would be useful. Similarly, a more detailed description of the signaling pathways (and better figures than Fig. 1), especially, brief characterisation of key signaling proteins affected by miRs would help the reader to understand these complex regulatory mechanisms. Biochemical activities of the most important proteins may be added to the Tables. Otherwise, a non-expert reader may find this paper a collection of weird abbreviations that do not make much sense.
  2. There are discrepancies between the text and Tables: some of the miRs (e.g. miR-3691-5p, miR-1307-3p, many others) are mentioned only in the text, while others (e.g. miR-93, miR-133b etc.) only in the Tables. c-Jun, an important regulator of the cell cycle and oncoprotein is mentioned in the text, but not listed in Table 4.
  3. There are some unexpected and unexplained players in the signaling pathways. What is the role of the tumor suppressor poritein WT1 in Wnt signaling (Table 3) and in liver cells? PTEN is a key tumor suppressor in the PI3K pathway, but not in RAS/MAPK signaling (Table 4).
  4. Figure 2 is very confusing and is not explained in the text at all. Many of the protein connections indicated are misleading or even incorrect (e.g. p21 regulating p53 /it is the other way around/; MDM2 both stimulating /?!/ and inhibiting p53; PTEN directly interacting with p53, etc.). This figure should be fundamentally revised and explained in the text.

Author Response

Reviewer 3

This manuscript is a review of microRNAs (miRs) potentially involved in liver carcinogenesis. Such RNAs are classified into two main categories: oncomiRs downregulate the expression of specific tumor suppressor proteins and are overexpressed in tumor cells; and tumor suppressor miRs that knock down the expression of specific oncoproteins and have reduced expression in cancer cells. The authors discuss some of the miRs that affect key signaling pathways involved in the regulation of cell division, survival, migration, invasion, metastasis etc., pathways that are implicated in liver carcinogenesis. MicroRNA-regulated signaling pathways covered by the review are the PI3K-, Hippo-, Wnt-, RAS/ERK- and p53-pathways. The review appears to be based on literature search (research or clinical contribution by the authors are not known by this reviewer). The manuscript is written in clear, polished language. It discusses a field of medical science of outstanding importance.

Response: Thank you for your positive and encouraging feedback. We appreciate your recognition of the clarity, relevance, and importance of the topic addressed in our manuscript.

     This reviewer, however, has several major problems with the manuscript:

  1. The audience targeted by this review are presumably medical personnel, many of them not having up-to-date knowledge in molecular biology and signal transduction. Therefore, a brief description of miR metabolism (transcription, processing etc.) and mechanism of action (perhaps with a simple figure) would be useful. Similarly, a more detailed description of the signaling pathways (and better figures than Fig. 1), especially, brief characterisation of key signaling proteins affected by miRs would help the reader to understand these complex regulatory mechanisms. Biochemical activities of the most important proteins may be added to the Tables. Otherwise, a non-expert reader may find this paper a collection of weird abbreviations that do not make much sense.

Response: We thank the reviewer for this valuable suggestion. In response, we have added a new section briefly describing miRNA biogenesis, processing, and mechanism of action, to help readers less familiar with molecular biology. We have also prepared a new, simplified figure to enhance clarity. We also modified the last figure 1 to make it more understandable. Furthermore, we have expanded the descriptions of the signaling pathways and included brief characterizations of the key signaling proteins regulated by miRNAs. The biochemical activities of the proteins have been added in tables. We believe these additions will make the manuscript more accessible to non-expert readers.

  1. There are discrepancies between the text and Tables: some of the miRs (e.g. miR-3691-5p, miR-1307-3p, many others) are mentioned only in the text, while others (e.g. miR-93, miR-133b etc.) only in the Tables. c-Jun, an important regulator of the cell cycle and oncoprotein is mentioned in the text, but not listed in Table 4.

Response: Thank you for pointing this out. We have now carefully cross-checked the text and Tables to ensure consistency. All microRNAs are now uniformly represented across both the text and the corresponding Tables. In addition, c-Jun has been added to Table 4 to reflect its role as an important regulator of the cell cycle and oncoprotein.

  1. There are some unexpected and unexplained players in the signaling pathways. What is the role of the tumor suppressor poritein WT1 in Wnt signaling (Table 3) and in liver cells? PTEN is a key tumor suppressor in the PI3K pathway, but not in RAS/MAPK signaling (Table 4).

Response: We thank the reviewer for this observation. We have revised the manuscript to clarify the role of WT1 in Wnt signaling and in liver cells. With respect to PTEN, we agree that its canonical role is as a negative regulator of the PI3K/AKT pathway, by dephosphorylating PIP3. It does not directly regulate RAS/MAPK signaling, and we have revised Table 4 to clarify this distinction.

  1. Figure 2 is very confusing and is not explained in the text at all. Many of the protein connections indicated are misleading or even incorrect (e.g. p21 regulating p53 /it is the other way around/; MDM2 both stimulating /?!/ and inhibiting p53; PTEN directly interacting with p53, etc.). This figure should be fundamentally revised and explained in the text.

Response: We appreciate your valuable comments. Figure 2 has been revised to correct misleading interactions (e.g., p53–p21, p53–MDM2) and improve clarity. We also modified the text to explain the figure for better understanding.

Round 2

Reviewer 1 Report

Comments and Suggestions for Authors

The authors addressed most of my concerns, I therefore recommend this manuscript for publication on IJMS

Reviewer 3 Report

Comments and Suggestions for Authors

     The extensive revision improved the quality of the manuscript. The critical comments (including the revision of the figures and the tables) have been properly addressed.